# Effect of Cocoa Beverage and Dark Chocolate Consumption on Blood Pressure in Those with Normal and Elevated Blood Pressure: A Systematic Review and Meta-Analysis

**DOI:** 10.3390/foods11131962

**Published:** 2022-07-01

**Authors:** Isaac Amoah, Jia Jiet Lim, Emmanuel Ofori Osei, Michael Arthur, Phyllis Tawiah, Ibok Nsa Oduro, Margaret Saka Aduama-Larbi, Samuel Tetteh Lowor, Elaine Rush

**Affiliations:** 1Department of Biochemistry and Biotechnology, Kwame Nkrumah University of Science and Technology, Kumasi 00233, Ghana; eoosei9@st.knust.edu.gh (E.O.O.); maarthur2@st.knust.edu.gh (M.A.); 2Human Nutrition Unit, School of Biological Sciences, University of Auckland, Auckland 1024, New Zealand; 3Department of Medicine, School of Medicine and Dentistry, College of Health Sciences, Kwame Nkrumah University of Science and Technology, Kumasi 00233, Ghana; phytawiah@knust.edu.gh; 4Department of Food Science and Technology, Kwame Nkrumah University of Science and Technology, Kumasi 00233, Ghana; inoduro.sci@knust.edu.gh; 5Cocoa Research Institute of Ghana, Akim-Tafo P.O. Box 8, Ghana; margaret.acheampong@crig.org.gh (M.S.A.-L.); slowor2@crig.org.gh (S.T.L.); 6Faculty of Health and Environmental Studies, Auckland University of Technology, Auckland 1142, New Zealand; elaine.rush@aut.ac.nz; 7Riddet Centre of Research Excellence, Palmerston North 0632, New Zealand

**Keywords:** cocoa powder, systolic blood pressure, diastolic blood pressure, polyphenol, flavanol, epicatechin

## Abstract

Cocoa is a major dietary source of polyphenols, including flavanols, which have been associated with reduced blood pressure (BP). While earlier systematic reviews and meta-analyses have shown significant effects of cocoa consumption on systolic BP, limitations include small sample sizes and study heterogeneity. Questions regarding food matrix and dose of polyphenols, flavanols, or epicatechins remain. This systematic review and meta-analysis aimed to investigate the effects of ≥2 weeks of cocoa consumption as a beverage or dark chocolate in those with normal or elevated (< or ≥130 mmHg) systolic BP measured in the fasted state or over 24-h. A systematic search conducted on PubMed and Cochrane Library databases up to 26 February 2022 yielded 31 suitable articles. Independent of baseline BP, cocoa consumption for ≥2 weeks was associated with reductions in systolic and diastolic BP (*p* < 0.05, all). Compared with cocoa, chocolate lowered the weighted mean of resting systolic BP (−3.94 mmHg, 95% CI [−5.71, −2.18]) more than cocoa beverage (−1.54 mmHg, 95% CI [−3.08, 0.01]). When the daily dose of flavanols was ≥900 mg or of epicatechin ≥100 mg, the effect was greater. Future, adequately powered studies are required to determine the optimal dose for a clinically significant effect.

## 1. Introduction

Cardiovascular diseases (CVDs) involve abnormalities of the heart and blood vessels, including coronary heart disease, peripheral arterial disease, and cerebrovascular disease [1]. Globally, CVD contributed to about 32% of deaths in 2019 [1]. Cerebrovascular accidents and ischaemic heart attacks are among the five main causes of death attributed to non-communicable diseases worldwide [2]. Elevated blood pressure (BP) and hypertension are the strongest risk factors linked to the causation of cardiovascular disease [3].

In adults, the international cut-off for hypertension is systolic BP ≥ 140 mmHg and diastolic BP ≥ 90 mmHg [4]. However, it has been proposed that the cut-off be reduced to 120–130 mmHg for systolic BP due to the clinical significance of systolic BP in predicting cardiovascular events [4]. For example, a 5 mm Hg decrease in systolic BP would be clinically significant in reducing CVD risks [5]. Therefore, systolic BP ≥ 130 mmHg is a treatable risk factor. Globally, a comprehensive analysis of data pooled from 1201 studies comprising 104 million individuals showed that the number of adults (aged 30–79 years) living with hypertension has doubled from 648 million adults in 1990 to 1.278 billion in 2019 [6], with the majority of the cases recorded in low and middle-income countries compared to higher-income countries [7,8]. In 2015, 7.45 million deaths that occurred in low and middle-income countries were attributed to systolic BP greater than 115 mmHg [4], with an increased incidence in sub-Saharan Africa. In 2021, despite improved treatments, the control of BP in sub-Saharan Africa and Oceania had not improved [6]. Risk factors, including obesity, increased sodium intake, alcohol consumption, lack of physical activity, and consumption of highly refined energy dense-nutrient poor foods, contribute to the increase in the prevalence of hypertension [8].

Cocoa bean products, including cocoa powder and dark chocolate, are foods that contain variable quantities of naturally-occurring polyphenols, most notably flavanols, including epicatechins [9,10]. It has been posited that an inverse association between the intake of flavanol-rich cocoa products and BP exists [11]. Flavanols antagonize the angiotensin-converting enzyme (ACE) and activate the enzyme nitric oxide synthase, resulting in increased nitric oxide synthesis [12]. The production of nitric oxide is associated with vasodilation [12] and reduction in BP. Hence, the consumption of cocoa, in addition to a diverse diet, can be promoted if sustainable production and availability of cocoa are possible and health effects are validated.

Several studies using animal models have established the bioavailability and anti-hypertensive effects associated with the intake of cocoa polyphenols. For example, in uninephrectomized hypertensive rats fed for 4 weeks with cocoa feed alone or in combination with an 8% salt diet, the authors reported reductions in both systolic and diastolic BP when compared to uninephrectomized rats fed normal feed [13]. In young spontaneously hypertensive rats, the administration of epicatechin for a two-week period resulted in a significant reduction in systolic BP [14]. Cienfuegos–Jovellanos et al. [15] administered a single dose of polyphenol-rich cocoa powder to male spontaneously hypertensive rats and reported that up to a dose of 300 mg/kg, there was a dose-dependent reduction in systolic BP in a manner similar to 50 mg/kg of Captopril (a known anti-hypertensive drug). The diastolic BP was maximally reduced over 24-h by the administration of 50 and 100 mg/kg of the polyphenol-rich cocoa powder. Maximum effects were achieved 4 h post-dose.

Cocoa is widely cultivated in West African countries, including Ghana, Ivory Coast, and Nigeria. In 2019, Ghana and Ivory Coast produced 811,700 and 2,180,000 tons of cocoa beans, respectively, which is more than 50% of the global cocoa supply [16]. After removing the husk, dried cocoa beans are ground into a powder, which is a major ingredient of dark chocolate. The Kuna Indians in the Caribbean consume large quantities of flavanol-rich cocoa powder and have very low BP and high longevity [17], which led to the initial proposal of a beneficial association between cocoa flavanols and cardiovascular health. Nowadays, chocolate is widely consumed in Western countries; its consumption is as high as 8.8 kg per capita in Switzerland, approximately 24 g/daily per person [18]. Chocolate is formulated with between 10 and 90% cocoa solids with sugar, flavors, milk, and emulsifiers added [19,20]. Hence, the health effect of cocoa consumed in the form of chocolate may depend on the formulation.

Findings from earlier reviews [20,21,22] have supported the overall effect of cocoa in the reduction of BP despite the large between-study heterogeneity, with many individual studies reporting null findings. We hypothesize that this may be due to inconsistencies surrounding the differences in participant characteristics, pre-intervention BP, the dose of cocoa or dark chocolate administered, and the food matrices. For example, Ried and colleagues [21], in their systematic review and meta-analysis, reported that even though the consumption of cocoa and dark chocolate improved BP, the effect was more prominent in people living with hypertension and pre-hypertension than those with normotension. Vlachojannis et al. [20] posited that the threshold daily intake of flavanol for improving BP was 900 mg, equivalent to the daily consumption of 100 to 500 g of chocolate, which was unrealistically high. The same authors further indicated that 100 mg epicatechin (one of the flavanols), equivalent to 50 to 200 g of chocolate, can reliably improve flow-mediated dilation (FMD) but not BP. In contrast, Hooper et al. [22] reported that the intake of 50 mg/day of epicatechin from cocoa or chocolate was sufficient to lower systolic and diastolic BP. These proposed thresholds have not been verified in a systematic review and meta-analysis. Therefore, this systematic review and meta-analysis aimed to clarify the effects of cocoa consumption in individuals with normal and elevated BP, whether cocoa and chocolate differentially modify BP, and to verify the threshold concentration of flavanol and epicatechin associated with a reduction in BP.

## 2. Methods

### 2.1. Literature Search Strategy

The systematic review was conducted in accordance with the guidelines of the recently updated Preferred Reporting Items for Systematic Reviews and Meta-Analysis (PRISMA) statement [23]. A systematic search for all published articles on cocoa and chocolate intake and its effect on BP was conducted on the databases PubMed and Cochrane Library on 26 February 2022. These two databases were chosen because they focus on “food and health”, and the Cochrane Library database records papers from Embase, the Cumulative Index to Nursing and Allied Health Literature (CINAHL), and the International Clinical Trials Registry Platform (ICTRP), and CT.gov. The search terms employed were “(cocoa OR cacao OR chocolate) AND (blood pressure OR pre-hypertension OR prehypertension OR pre-hypertensive OR prehypertensive OR hypertension OR hypertensive OR normotension OR normotensive OR cardiovascular OR vascular)”.

During the article search and retrieval, there was no use of filters or restriction of dates and language. Records obtained from PubMed and Cochrane Library were imported to the reference manager, ENDNOTE™ Library. Duplicate records were removed using the ENDNOTE™ Library. The final articles selected for the systematic review were critically checked to ensure they met the inclusion and exclusion criteria.

### 2.2. Selection of Studies

#### 2.2.1. Eligibility Criteria

The inclusion criteria adopted for the selection of records for the systematic review include: (i) articles published must be original research; (ii) articles must be written in English with its full text readily accessible; (iii) articles must report BP at baseline and at post-intervention or the change in BP at the post-intervention; (iv) study duration must be equal to or greater than two weeks; (v) study must compare a cocoa product as a treatment against a non-cocoa placebo control or a cocoa product that has a negligible content of polyphenols; (vi) adult population (age equal or greater than 18 years old).

#### 2.2.2. Exclusion Criteria

The exclusion criteria were: (i) studies that were published as review articles, commentaries, letters to editors, and conference proceedings; (ii) studies that use cocoa or chocolate intervention along with other lifestyle interventions, such as exercise or weight-loss intervention; (iii) studies that use cocoa or chocolate supplemented with other nutrients; (iv) the study population had significant comorbidities, such as type-2 diabetes mellitus and CVD.

#### 2.2.3. Data Extraction

Study characteristics extracted were the year of publication, country of study, age and body mass index (BMI) of participants, sample size, study design, baseline BP, intervention duration (weeks), food matrix, servings per day, daily dose of polyphenol, flavanol, or epicatechin, and study outcomes. The primary outcome of this systematic review and meta-analysis is resting systolic BP, resting diastolic BP, 24-h systolic BP, and 24-h diastolic BP. Resting BP was measured in the fasted state, whereas 24-h BP was measured using an ambulatory BP-measuring device. The study population was categorized as normal BP (mean systolic BP < 130 mmHg) or elevated BP (mean systolic BP ≥ 130 mmHg).

The primary comparison of this systematic review was to compare the effect of cocoa consumption in a population with elevated BP against the effect of cocoa consumption on BP in a population with normal BP. The other prespecified comparisons included comparing the effect of study duration, food matrices (beverage, chocolate, combination of beverage and chocolate, and cocoa extract in the form of tablets or capsules), daily dose of polyphenol, daily dose of flavanol, and daily dose of epicatechin. Three independent authors, including I.A., E.O.O., and M.A., carried out the article search. In instances where there was a lack of agreement among I.A., E.O.O., and M.A. on the inclusion or exclusion of identified records, a consensus was reached by further discussion between I.A., E.O.O., M.A., and J.J.L.

#### 2.2.4. Meta-Analysis

Meta-analysis was conducted using the Review Manager (Version 5.4.1, Cochrane Collaboration, Oxford, England). The mean difference (±standard deviation of the difference, SD_diff_) in BP at post-intervention relative to pre-intervention was extracted to calculate the pooled effect in the meta-analysis. Where the mean difference and SD_diff_ in BP were not reported, we calculated the mean difference by using post-intervention mean minus pre-intervention mean and imputed SD_diff_ by using the equation recommended in the Cochrane Handbook for Systematic Reviews of Interventions Version 6.2 [24]
$${\mathrm{SD}}_{\mathrm{diff}}=\sqrt{SD_{post}{}^{2}+SD_{pre}{}^{2}-\left(2\times Corr\times SD_{post}\times SD_{pre}\right)},$$

*SD_post_* is the SD of post-intervention mean, *SD_pre_* is the SD of pre-intervention mean, *Corr* is the correlation between *SD_post_* and *SD_pre_*, which was assumed to be 0.5 based on Follmann et al. [25].

When the unit of variance was reported as standard error or confidence interval, conversion into SD was conducted in accordance with the Cochrane guidelines. When the data was only available in the form of graphs, a pdf-measurement tool (Adobe Acrobat Pro DC, Burlington, NJ, USA) was used to estimate the values [24].

In the treatment groups, cocoa was consumed in the form of beverages, dark chocolate, and cocoa extract. In the control groups, either there was no cocoa consumption, or placebo or cocoa with a negligible amount of flavanol were consumed in place of cocoa. In studies where there were more than two intervention groups, we avoided double-counting the same participant by selecting the most relevant groups as treatment and control. In the case of a dose-rising study, the highest dose (treatment) was compared against the lowest dose (control), which had a negligible amount of flavanols. In studies where two different populations (e.g., hypertensive population and normotensive population) were reported, the two different populations were included as two separate treatment comparisons in the meta-analysis.

The primary objective of this meta-analysis was to compare the effect of cocoa consumption on BP in individuals living with normal BP (mean systolic BP < 130 mm Hg) and elevated BP (mean BP ≥ 130 mm Hg) using subgroup analysis. Other prespecified subgroup analyses were also conducted to understand whether the effect of cocoa on improving BP may differ (i) among different food matrices (beverage, chocolate, vs. cocoa extract), (ii) between studies of different duration, (iii) between different daily doses of polyphenols, (iv) between different daily doses of flavanols (<900 mg vs. ≥900 mg daily as proposed by Vlachojannis et al. [20]), and (v) between different daily doses of epicatechin (<100 mg vs. ≥100 mg daily, similarly proposed by Vlachojannis et al. [20]).

The overall effect size was computed as weighted mean difference (WMD) with a 95% confidence interval (CI), using a random-effect model, and presented as forest plots. Between-study heterogeneity and between-subgroup heterogeneity were computed as *I*^2^, whereby *I*^2^ of ≤40%, 30–60%, 50–90%, and 75–100% represent low, moderate, substantial, and considerable heterogeneity, respectively. We also assessed the robustness of the results by conducting sensitivity analysis using the leave-one-out method [26] and removing ambiguous data.

#### 2.2.5. Risk of Bias

An assessment of the risk of bias was carried out for each study included in this systematic review. This followed the Cochrane risk of bias tool for randomized controlled trials based on the parameters: random sequence generation, allocation concealment, blinding of participants or study personnel, blinding of outcome assessment, incomplete outcome data, selective reporting, and other bias [24].

## 3. Results

A total of 1659 records were retrieved from the databases PubMed and Cochrane Library. Following the removal of 232 duplicate records, a total of 1427 records were obtained. The 1427 records were screened for their titles and abstracts, resulting in 75 records that were assessed for eligibility by applying the inclusion and exclusion criteria. A total of 28 study reports were identified as eligible. In addition, three other study reports were identified from previous systematic reviews. A total of 31 study reports were included in this systematic review and meta-analysis (Figure 1). The detailed study and participant characteristics are summarized in (Table 1).

Of the 31 study reports, 36 unique treatment comparisons were identified for the meta-analysis, as some studies reported multiple treatment comparisons. The study population in two-thirds of the treatment comparisons had normal baseline BP (*n* = 23, 64%), and one-third had elevated baseline BP (*n* = 13, 36%) (Figure 2).

In the normal baseline BP group (Figure 2), 17 treatments (74%) were of younger age (mean age ≤ 45 years old), 4 treatments (17%) were of middle-to-older age (mean age > 45 years), and 2 treatments (9%) in the Almoosawi et al. [27] study did not report the age of participants. Furthermore, the sample population in the majority of treatments (*n* = 21, 91%) were generally normal-to-overweight (mean BMI ≤ 30 kg/m^2^), 1 treatment (4%) reported in the Almoosawi et al. [27] study was comprised of individuals with mainly overweight and obesity (BMI > 25 kg/m^2^), 1 other treatment (4%) reported in the Davison et al. [32] study was comprised of mainly obese individuals (BMI > 30 kg/m^2^). The majority of the treatments (*n* = 18, 78%) also had an intervention duration of ≤4 weeks, with only *n* = 5 treatments (22%) having an intervention duration of >4 weeks. Cocoa was delivered as a beverage in 10 treatments (43%), as chocolate in 8 treatments (34%), as a combination of beverage plus chocolate in 2 treatments (9%), and as tablets/capsules in 3 treatments (13%). A total of 11 of the 23 treatments (47%) reported the daily dose of polyphenols, whereby the daily dose of polyphenols was <500 mg in 4 treatments (17%) and ≥500 mg in the other 7 treatments (30%). Additionally, 11 of the 23 treatments (47%) also reported the daily dose of flavanols, whereby the daily dose of flavanols was <900 mg in 7 treatments (30%), and ≥900 mg in the other 4 treatments (17%). Additionally, 11 of the 23 treatments (47%) reported the daily dose of epicatechin, whereby daily dose of epicatechin was <100 mg in 8 treatments (35%) and ≥100 mg in the other 3 treatments (13%).

In contrast to the normal BP group, the sample population of the elevated BP group (Figure 2), was middle-to-older age (mean age > 45 years old) in the majority of treatments (*n* = 10, 77%). Similar to the normal BP group, the study population was primarily normal-to-overweight (mean BMI ≤ 30 kg/m^2^) in the majority of treatments (*n* = 10, 69%) and was primarily obese (mean BMI >30 kg/m^2^) in only 3 treatments (23%). Approximately half of the treatments (*n* = 6, 46%) had an intervention duration of ≤4 weeks, and the other half (*n* = 7, 54%) had an intervention duration of >4 weeks. Cocoa was delivered as a beverage in 5 treatments (38%), as chocolate in 6 treatments (46%), and as a combination of beverage plus chocolate in 2 treatments (15%). No treatments were delivered as tablets/capsules. Additionally, 5 of the 13 treatments (38%) reported the daily dose of polyphenols, whereby the daily dose of polyphenols was <500 mg in 1 treatment (8%) and ≥500 mg in the other 4 treatments (31%). Additionally, 9 of the 13 treatments (69%) also reported the daily dose of flavanols, whereby the daily dose of flavanols was <900 mg in 6 treatments (46%) and ≥900 mg in the other 3 treatments (23%). Additionally, 8 of the 13 treatments (62%) reported the daily dose of epicatechin, whereby the daily dose of epicatechin was <100 mg in 4 treatments (31%) and ≥100 mg in the other 4 treatments (31%).

### 3.1. Risk of Bias Assessment

The risk of bias assessment is summarized in Figure 3. Approximately half of the studies did not report random sequence generation and allocation concealment, resulting in an unclear risk of bias. Blinding of participants and personnel was not undertaken in 10 studies. Blinding was difficult to maintain when the intervention product was provided in the form of dark vs. white chocolate. In contrast, blinding was better maintained when the intervention product was provided as high-flavanol cocoa vs. cocoa with a negligible amount of flavanols. Detection bias was low since the measurement of resting BP was carried out by trained researchers using standardized methods and conditions. Approximately a quarter of studies had an attrition rate greater than 15%; hence are regarded as a high risk of attrition bias. Most studies had low reporting bias as the studies reported BP measurement for each occasion when BP was measured. However, two studies had unclear reporting bias as the baseline BP measurement was not clear. We did not detect any other bias of importance. However, it was noteworthy that the study population in the Fraga [34] study was active soccer players, regarded as an unclear risk of bias. In general, the study reports were of low risk of bias.

### 3.2. Meta-Analysis

The pooled effect of cocoa consumption on BP and the outcome of prespecified subgroup analysis are discussed in the proceeding sections.

#### 3.2.1. Effect of Cocoa Consumption on Resting BP

Meta-analysis of 34 treatment comparisons showed that resting systolic (Figure 4) and diastolic BP (Figure 5) were significantly lower (systolic BP, WMD = −1.87, 95% CI [−3.08, −0.65] mmHg, *p* = 0.003; diastolic BP, WMD = −1.21, 95% CI [−2.08, −0.35] mmHg, *p* < 0.001) following the consumption of cocoa compared to control. Subgroup analysis showed that the BP-lowering effect of cocoa was not significantly different between individuals with normal and elevated BP (systolic BP, *p* = 0.68; diastolic BP, *p* = 0.68). Notably, there was substantial between-study heterogeneity in the effect size in the elevated BP subgroup (systolic, *I*^2^ = 78%; diastolic, *I*^2^ = 68%), but low between-study heterogeneity in the effect size in the normal BP subgroup (systolic, *I*^2^ = 7%; diastolic, *I*^2^ = 6%). Notably, Taubert et al. [54] reported an unusually small SD in BP relative to other studies; hence the data in this study was flagged as ambiguous (Figure 4 and Figure 5). Sensitivity analysis showed that removing this data from the meta-analysis neither changed the significance of the pooled effect (systolic BP, WMD = −1.67, 95% CI [−3.08 −0.27] mmHg, *p* = 0.02; diastolic BP, WMD = −1.32, 95% CI [−2.22, −0.41] mmHg, *p* = 0.004), nor lower the heterogeneity of the elevated BP subgroup (systolic BP, *I*^2^ = 80%; diastolic BP, *I*^2^ = 71%). Overall, cocoa consumption lowered resting systolic and diastolic BP in populations with normal and elevated BP, though the pooled effect size was small.

#### 3.2.2. Effect of Cocoa Consumption on 24-h BP

Meta-analysis of 13 treatment comparisons showed that 24-h systolic (Figure 6) and diastolic BP (Figure 7) were significantly lower (systolic BP, WMD = −2.64, 95% CI [−5.02, −0.26] mmHg, *p* = 0.03; diastolic BP, WMD = −2.21, 95% CI [−4.04, −0.38] mmHg, *p* = 0.02) following the consumption of cocoa compared to control. Similar to the resting BP outcomes, subgroup analysis showed that the BP-lowering effect of cocoa was not significantly different between individuals with normal and elevated BP (systolic BP, *p* = 0.95; diastolic BP, *p* = 0.76). There was substantial to considerable heterogeneity in the effect size in the elevated BP subgroup (systolic, *I*^2^ = 81%; diastolic, *I*^2^ = 82%), but lower heterogeneity in the effect size in the normal BP subgroup (systolic, *I*^2^ = 27%; diastolic, *I*^2^ = 59%). Overall, cocoa consumption lowered 24-h systolic and diastolic BP in populations with normal and elevated BP.

#### 3.2.3. Effect of Cocoa Consumption on BP based on Intervention Duration

The median intervention duration was 4 weeks, ranging between 2 and 26 weeks. We compared the effect of cocoa on BP between interventions of ≤4 weeks and >4 weeks. Subgroup analysis (Table 2) showed that the effect of cocoa in lowering resting systolic BP was not significantly different between subgroups (*p* = 0.22). Unexpectedly, there was a greater reduction in diastolic BP when the intervention duration was ≤4 weeks (*p* = 0.03). Although sensitivity analysis showed that the subgroup difference in diastolic BP was also robust to the leave-one-out method, grouping by intervention duration did not help with explaining the overall heterogeneity between studies for both resting systolic and diastolic BP outcomes.

Similarly, the effect of cocoa in lowering 24-h systolic BP was not significantly different between subgroups (*p* = 0.15), but its effect on lowering 24-h diastolic BP was significantly greater when intervention duration was ≤4 weeks (*p* = 0.02). Sensitivity analysis showed that the subgroup difference in 24-h diastolic BP was not robust to the leave-one-out method. Subgroup analysis showed that there was substantial heterogeneity between studies with duration ≤ 4 weeks but no heterogeneity between studies with intervention duration > 4 weeks. Hence, there was low certainty of evidence that the effect of cocoa in improving 24-h diastolic BP was different between interventions of ≤4 weeks and >4 weeks. Overall, the effect of cocoa consumption on BP was not significantly improved with a longer intervention duration, whereby an intervention duration of two weeks was sufficient to demonstrate the effect of cocoa in improving BP outcomes.

#### 3.2.4. Effect of Food Matrices

Based on the studies identified, cocoa could be delivered in the format of beverage, chocolate, a mixture of beverage plus chocolate, and tablets or capsules. Interestingly, subgroup analysis (Table 2) comparing these food matrices showed a statistically significant difference in the effect size between various food matrices for resting systolic BP (*p* = 0.002), resting diastolic BP (*p* = 0.003), and 24-h diastolic BP (*p* = 0.04), but not 24-h systolic BP (*p* = 0.10). Interestingly, the effect of cocoa in lowering resting systolic BP was significantly greater when delivered in the format of chocolate when compared to cocoa beverage (Beverage, WMD = −1.54, 95% CI [−3.08, 0.01] mmHg; Chocolate, WMD = −3.94, 95% CI [−5.71, −2.18]; subgroup differences, *p* = 0.04), but not resting diastolic BP (*p* = 0.08), 24-h systolic BP (*p* = 0.15), and 24-h diastolic BP (*p* = 0.40). However, sensitivity analysis showed that the greater effect of chocolate in lowering resting systolic BP relative to cocoa beverage was not robust to the leave-one-out method. Furthermore, the heterogeneity between studies was higher when cocoa was delivered in the form of chocolate compared to a beverage. Therefore, this finding needs to be interpreted with caution.

#### 3.2.5. Effect of Daily Dose of Polyphenols

The median daily dose of polyphenols was 500 mg, ranging between 30–1008 mg. We compared the effect of cocoa when the daily dose of polyphenols was <500 mg against ≥500 mg. This subgroup analysis (Table 2) consisted of 15 treatment comparisons for resting systolic and diastolic BP outcomes. The subgroup analysis showed that the effect of cocoa on lowering resting systolic and diastolic BP was not significantly different between the two subgroups (systolic, *p* = 0.28; diastolic, *p* = 0.93). As previously described, we identified Taubert et al. [54] as ambiguous data. Removing Taubert et al. [54] data in the sensitivity analysis did not alter the statistical significance of the subgroup differences. Subgroup analysis was not conducted for 24-h systolic and diastolic BP outcomes since there were only 2 treatments that reported a daily dose of polyphenols.

#### 3.2.6. Effect of Daily Dose of Flavanols

The median daily dose of flavanols was 514 mg, ranging between 45.3–2000 mg. However, we compared the effect of cocoa when delivered as <900 mg against ≥ 900 mg daily dose of flavanols, following the cut-off proposed by Vlachojannis et al. [20]. This subgroup analysis (Table 2) consisted of 17 treatment comparisons for resting systolic and diastolic BP outcomes and 12 treatment comparisons for 24-h systolic and diastolic BP outcomes. The subgroup analysis showed that the effect of cocoa on lowering BP was not significantly different between the 2 subgroups for resting systolic BP (*p* = 0.24), 24-h systolic BP (*p* = 0.85), and 24-h diastolic BP (*p* = 1.00), but was significantly different for resting diastolic BP (*p* = 0.04). However, we identified Grassi et al. [38] as potentially ambiguous data due to its unusually strong effect size in the <900 mg subgroup, although the daily dose of flavanols was among the lowest (88 mg/day) in the subgroup. Hence, a sensitivity analysis was performed following the removal of Grassi et al. [38] from the <900 mg flavanols subgroup. The analysis then showed that the effect of cocoa in lowering resting systolic BP was now significantly greater when the daily dose of flavanols was ≥900 mg (<900 mg, WMD = 1.14, 95%CI [−0.60, 2.88] mmHg; ≥900 mg, WMD = −2.88, 95% CI [−5.06, −0.70] mmHg; subgroup differences, *p* = 0.005) and the heterogeneity (*I*^2^) in the <900 mg flavanols subgroup was lowered from 79% to 0%, confirming that Grassi et al. [38] was an unusual data contributed to the heterogeneity between studies in the <900 mg flavanols subgroup. Following the removal of the Grassi et al. [38] data, the effect of cocoa in lowering resting diastolic BP remained significantly greater when the daily dose of flavanols was ≥900 mg (<900 mg, WMD = 0.43, 95%CI [−0.67, 1.52] mmHg; ≥900 mg, WMD = −2.88, 95% CI [−4.43, −1.33] mmHg; subgroup differences, *p* < 0.001), and the heterogeneity (*I*^2^*)* in the <900 mg flavanols subgroup was also lowered from 64% to 0%. The effect of cocoa in lowering 24-h systolic and diastolic BP remained not significantly different between subgroups even after the removal of Grassi et al. [38] data. Overall, there was certainty in evidence that cocoa consumption may lower resting systolic and diastolic BP when the daily dose of flavonols in the cocoa was ≥900 mg when excluding Grassi et al. [38] data.

#### 3.2.7. Effect of Daily Dose of Epicatechin

The median daily dose of epicatechin was 46 mg, ranging between 4–220 mg. However, we compared the effect of cocoa when delivered as <100 mg against ≥100 mg daily dose of epicatechin, following the cut-off proposed by Vlachojannis et al. [20]. This subgroup analysis (Table 2) consisted of 19 treatment comparisons for resting systolic and diastolic BP outcomes and 10 treatment comparisons for 24-h systolic and diastolic BP outcomes. The subgroup analysis showed that the effect of cocoa on lowering BP was not significantly different between 2 subgroups for resting systolic BP (*p* = 0.38), resting diastolic BP (*p* = 0.10), 24-h systolic BP (*p* = 0.79), and 24-h diastolic BP (*p* = 0.81). Similar to the sensitivity analysis conducted for flavanols, Grassi et al. [38] data was then removed. Consequently, the effect of cocoa in lowering resting systolic BP became significantly greater when the daily dose of epicatechin was ≥100 mg (<100 mg, WMD = −0.23, 95% CI [−1.82, 1.35] mmHg; ≥100 mg, WMD = −3.01, 95% CI [−4.58, −1.44] mmHg; subgroup differences, *p* = 0.01), and the heterogeneity (*I*^2^) in the <100 mg epicatechin subgroup was lowered from 78% to 4%, again confirming that Grassi et al. [38] data was unusual. The effect of cocoa in lowering resting diastolic BP also became significantly greater when the daily dose of epicatechin was ≥100 mg (<100 mg, WMD = −0.12, 95%CI [−1.19, 0.96] mmHg; ≥100 mg, WMD = −2.82, 95% CI [−4.18, −1.45] mmHg; subgroup differences, *p* = 0.002), and the heterogeneity (*I*^2^) in the <100 mg epicatechin subgroup was also lowered from 65% to 7%. The effect of cocoa in lowering 24-h systolic BP also became significantly greater in the ≥100 mg subgroup (<100 mg, WMD = 0.66, 95% CI [−1.43, 2.76] mmHg; ≥100 mg, WMD = −3.66, 95% CI [−5.68, −1.63] mmHg; subgroup differences, *p* = 0.004), but did not change the statistical outcome for 24-h diastolic BP. Overall, there was a certainty of evidence that cocoa consumption lowered resting systolic BP, resting diastolic BP, and 24-h diastolic BP when the daily dose of epicatechin was ≥100 mg, after excluding Grassi et al. [38] data.

## 4. Discussion

Meta-analyses conducted on the studies selected for this systematic review consistently showed that cocoa consumption was significantly associated with a reduction in BP in the order of 1 to 5 mmHg. This effect was observed when the BP was measured in the fasting state or for a 24-h period. The systolic and diastolic BP were significantly lower in the fasting state after cocoa was consumed, and this was independent of whether the treatment groups had normal BP or elevated BP (Figure 4 and Figure 5). In agreement with the BP outcome measured in the fasted state, cocoa consumption significantly lowered 24-h systolic and diastolic BP in populations with normal and elevated BP (Figure 6 and Figure 7). These findings are consistent with what others have reported in past systematic reviews and meta-analyses [21,22,58]. The authors reported that cocoa consumption was associated with significant reductions in systolic BP [21,58] and diastolic BP [21,22,58] in normotensive [58] and people living with pre-hypertension [21,58] and hypertension [21,58] and adult participants at any risk of CVD, but not critically ill [22]. The plausible rationale for the improved BP outcomes associated with cocoa consumption includes that cocoa powder is a rich source of polyphenol flavanol [59]. Flavanol increases the production of nitric oxide and enhances its bioavailability, reduces oxidative stress in vascular endothelium, and promotes blood flow [60,61]. This results in increased vasodilation and improves blood pressure outcomes [61]. Importantly, the overall pooled effect size in lowering BP is small (systolic BP, WMD = −1.87, 95% CI [−3.08, −0.65] mmHg, *p* = 0.003), and the lower CI did not meet the 5 mmHg reduction in systolic BP that would be considered as clinically significant. Nonetheless, our meta-analyses identified factors that may play a role in increasing the effectiveness of cocoa as part of treatment for lowering BP.

The food matrix affects food’s health-promoting properties. This is fundamentally due to issues of bioavailability and bioaccessibility of the specific nutrients of interest [62]. In the case of cocoa, the predominant bioactive compound of interest that has been purported for its BP-lowering properties is the polyphenolic compound flavanol [9,11]. There are other components in food that may either synergistically enhance the food’s bioactivity or impair it. Our meta-analysis showed that the effect of cocoa in lowering resting systolic BP was significantly greater when delivered in the format of chocolate when compared to cocoa beverage (Beverage, WMD = −1.54, 95% CI [−3.08, 0.01] mmHg; Chocolate, WMD = −3.94, 95% CI [−5.71, −2.18]; subgroup differences, *p* = 0.04), but not resting diastolic BP (*p* = 0.08), 24-h systolic BP (*p* = 0.15), and 24-h diastolic BP (*p* = 0.40). The enhanced bioactivity recorded for cocoa delivered particularly in the form of chocolate, could be attributed to the presence of nutrients such as fats which may increase the bioavailability of the polyphenol and slow digestion. Roura et al. [63] conducted a randomized controlled cross-over study to investigate the effect of cocoa polyphenol bioavailability in different food matrices. In that study, the participants took (i) 250 mL of whole milk (control), (ii) 40 g of cocoa powder dissolved in 250 mL of whole milk, and (iii) 40 g of cocoa powder dissolved with 250 mL of water. The authors posited that the addition of milk to cocoa powder does not impair the bioavailability of the polyphenols in cocoa [63]. However, chocolate products are very heterogenous, containing variable amounts of cocoa solids, sugar, and fat [19], reflecting the high between-study heterogeneity in our meta-analyses. Investigation into whether high-fat or high-sugar chocolate can counteract the effect of cocoa polyphenols in improving BP will be worth exploring. Furthermore, other cardiometabolic outcomes, such as blood glucose and lipid profile, associated with the consumption of chocolate vs. cocoa beverages should also be considered. We are aware that a study reported that endothelial function was not significantly improved following sugar-free cocoa beverages vs. sugar-sweetened cocoa beverages [64]. It must, however, be emphasized that although, based on this present meta-analysis, it appeared that the effect size of chocolate in reduction of BP was greater than cocoa beverage, the difference was not robust to sensitivity analysis. Moreover, no studies directly compared the BP-reducing effect following the consumption of cocoa beverage and chocolate that contains identical amount of flavanols. Such studies will be of high value to shed light on the effect of food matrices on the bioavailability and bioaccessibility of cocoa flavanols, eventually identifying the optimum food matrix and composition to deliver cocoa flavanol for optimum BP-lowering effect.

Regarding the duration of intervention, the meta-analysis showed that significant improvement in BP outcomes following cocoa consumption was not associated with a longer intervention duration. In fact, an intervention duration of 2 weeks was sufficient to demonstrate the effect of cocoa in improving BP outcomes. This observation could be impacted by factors including the dose and the matrix through which flavanol, the predominant bioactive compound in cocoa [12], is delivered. The meta-analysis showed that there was certainty in evidence that cocoa consumption may lower resting systolic and diastolic BP when the daily dose of flavonols in the cocoa was ≥900 mg (equivalent to 100–500 g chocolate) or epicatechin ≥ 100 mg (equivalent to 50–200 g chocolate), confirming Vlachojannis et al. [20] recommendations. To deliver this amount of cocoa flavanols or epicatechin, cocoa powder is preferred over chocolate as the former has a higher concentration of flavanols or epicatechin and is less energy-dense [19]. Puzzlingly, we also showed that cocoa delivered in the format of chocolate appeared to exert a greater physiological effect in lowering BP. Therefore, we proposed future research to investigate how food matrices affect the bioavailability and bioaccessibility of polyphenols and how to enrich chocolate with a greater amount of flavanols without causing excessive energy intake. The sample size of most studies was generally small, and thus the studies had insufficient power to detect small changes in BP. A 5 mmHg decrease in systolic BP was proposed to be clinically significant in reducing the CVD risks [5]. Based on our meta-analysis, the pooled SD_diff_ for systolic BP was 12.5 mmHg. We recommended that power calculation for future studies should base on this SD_diff_. Hence, to detect a 5 mmHg difference in systolic BP with 80% power requires a total sample size of *n* = 52 for a cross-over trial and *n* = 100 for each treatment group for a parallel trial. Consequently, it is now not surprising that many individual studies did not detect significant differences in BP between control and treatment groups.

Most of the studies were conducted in the USA (*n* = 11), Italy (*n* = 5), Spain (*n* = 4), UK (*n* = 3), Cote D’Ivoire (*n* = 2), Australia (*n* = 3), Germany (*n* = 3), Finland (*n* = 1), Mexico (*n* = 1), Japan (*n* = 2), Brazil (*n* = 1), Netherlands (*n* = 1), Saudi Arabia (*n* = 1), and Indonesia (*n* = 1). Ghana and Cote D’Ivoire produce about 60% of the global cocoa beans annually. Consequently, there is the expectation that these two countries will be the frontiers in cocoa research, particularly in the validation of the health-promoting effects associated with cocoa consumption. The validation of the health-promoting effects of cocoa through clinical trials is essential for attracting a market premium that could be used to market cocoa beans from these African countries. Interestingly, in this review, most of the studies were conducted in the USA, Italy, Spain, and UK. Clinical trials are generally expensive. In Africa, access to funding for research is a challenge compared to developing countries. These factors, coupled with an apparent publication bias, could possibly account for this observation.

There are several limitations in this systematic review and meta-analysis. First, we only sampled populations without comorbidities. Thus, the outcome of this meta-analysis cannot be generalized to individuals with existing comorbidities, such as diabetes or with a history of CVD. Second, the sample size of most of the included studies used for this systematic review and meta-analysis was small and was considered of moderate quality by a previous review [65]. Consequently, a health claim for cocoa flavanols to reduce BP could not be recommended [65], further supported by the small overall effect size of cocoa in lowering systolic BP shown in this systematic review and meta-analysis. In addition, other factors that may modify the BP-lowering effect of cocoa were not fully elucidated. In this systematic review and meta-analysis, we thoroughly investigated how study duration, food matrices, daily dose of polyphenols, daily dose of flavanols, and daily dose of epicatechin may affect the BP-lowering properties of cocoa. Other potential factors, including the number of servings per day, timing of intake, and intake with or without food, may be of interest for future investigations. Compliance with and feasibility of interventions longer in duration should also be investigated as BP reduction requires daily treatment.

## 5. Conclusions

The present systematic review and meta-analysis showed that the consumption of cocoa significantly reduced resting systolic and diastolic BP, and over a 24-h period in both individuals with normal and elevated BP in the order of 1 to 5 mmHg. However, the pooled effect size of cocoa in reducing BP could not be regarded as clinically significant. Factors including food matrices and doses of flavanols and epicatechins should be addressed as they may modify the effectiveness of cocoa in reducing BP. The use of chocolate as the delivery medium for flavanols in cocoa in the human gut may be more effective for reduction in BP compared to its beverage form but warrants future investigations. The effect of cocoa in lowering BP was generally greater when the daily dose of flavanols was ≥900 mg or epicatechin ≥100 mg. Future studies should investigate the effect of cocoa beverage vs. chocolate and the dose of flavanols or epicatechins in modifying the effect of cocoa in reducing BP with an adequately powered sample size.

## Figures and Tables

**Figure 1 foods-11-01962-f001:**
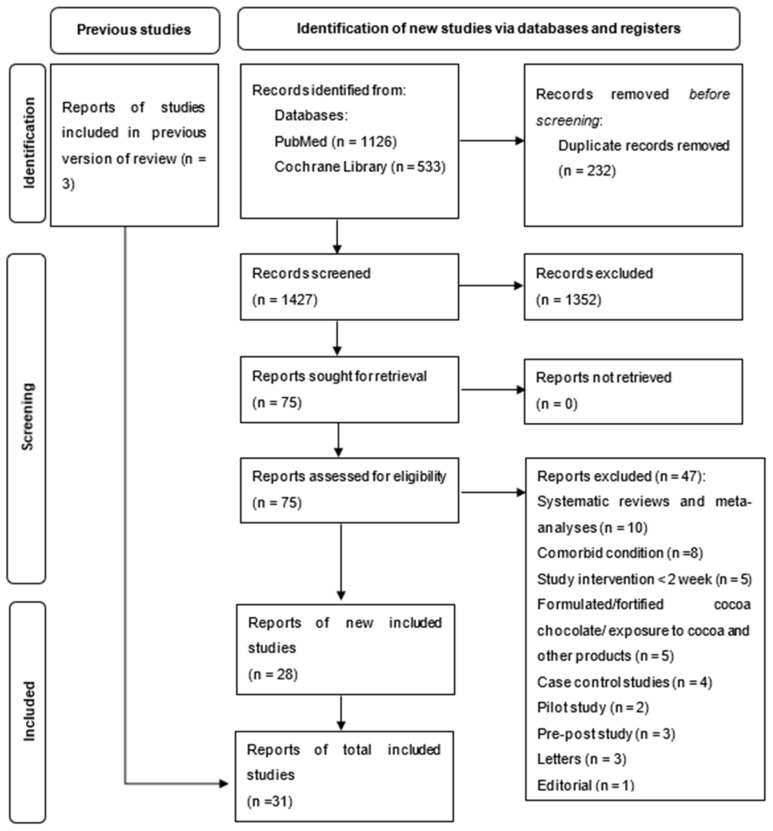
Flowchart summarizing studies evaluated and selected for the systematic review.

**Figure 2 foods-11-01962-f002:**
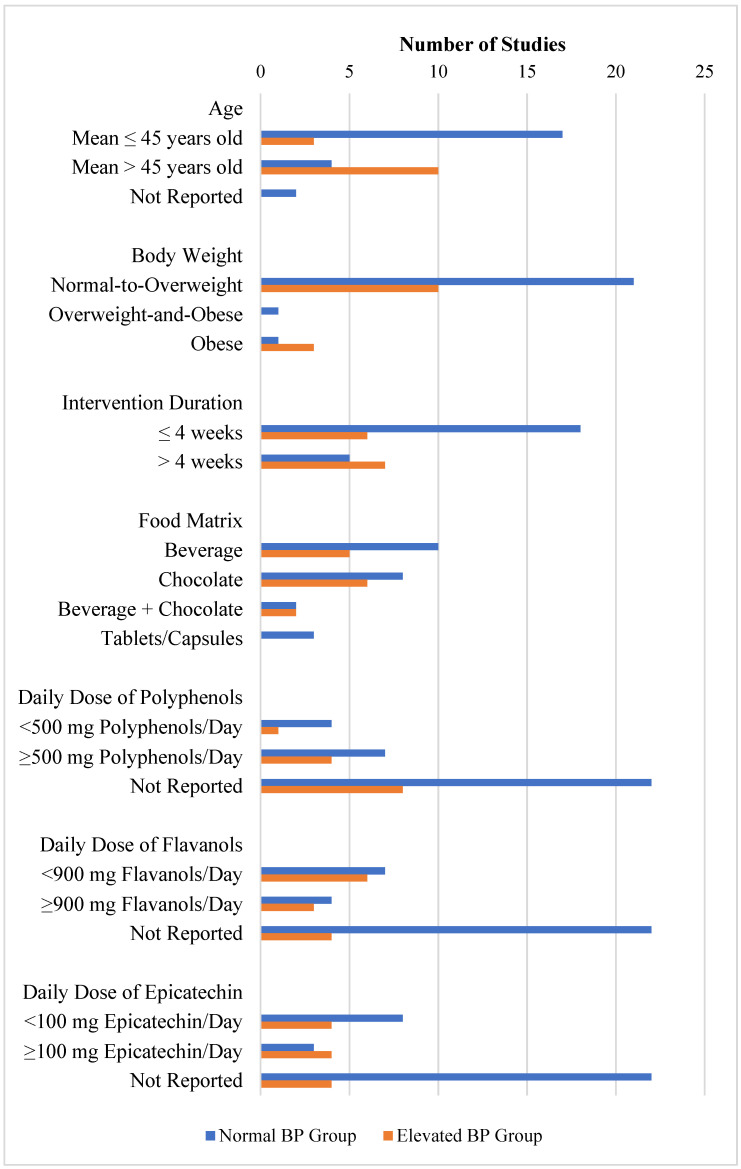
Summary of study characteristics grouped by pre-intervention BP.

**Figure 3 foods-11-01962-f003:**
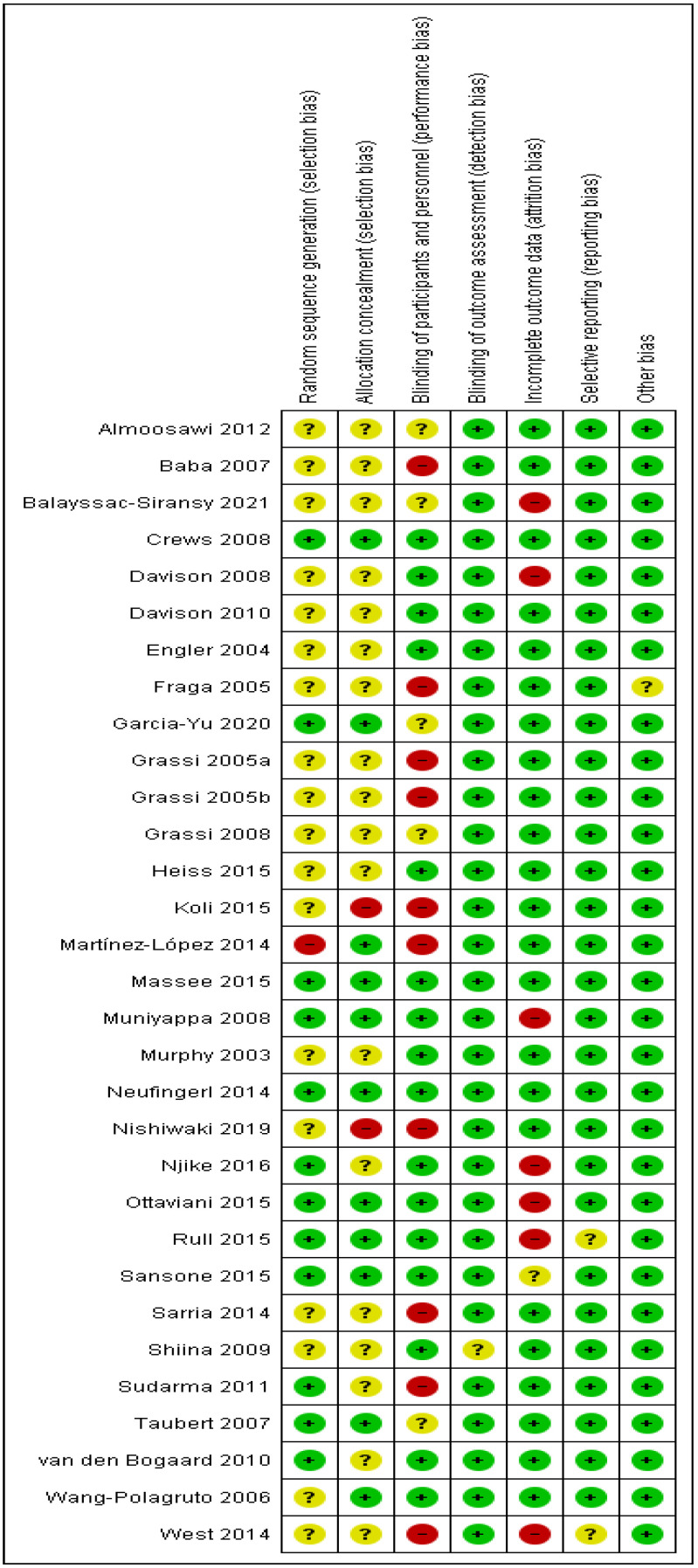
Risk of bias assessment for each study report. Symbols: ?, unclear risk of bias; +, low risk of bias; -, high risk of bias.

**Figure 4 foods-11-01962-f004:**
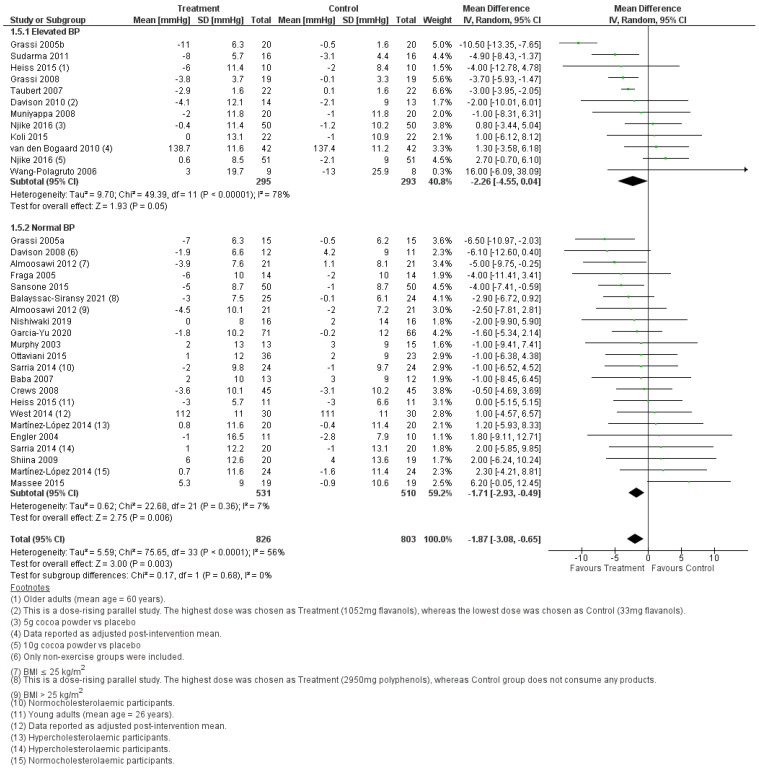
Forest plot showing the pooled effect of cocoa consumption on resting systolic BP.

**Figure 5 foods-11-01962-f005:**
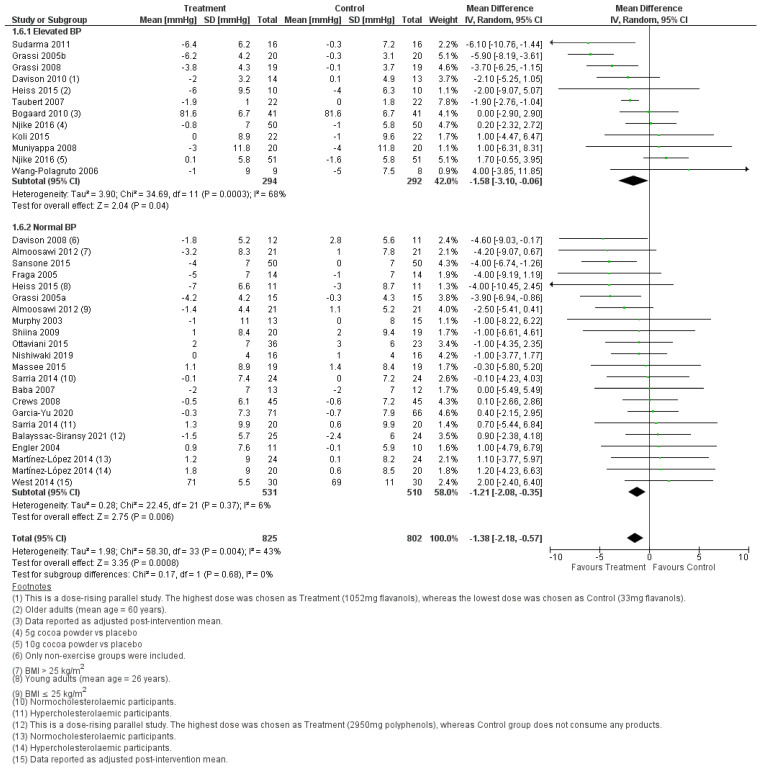
Forest plot showing the pooled effect of cocoa consumption on resting diastolic BP.

**Figure 6 foods-11-01962-f006:**
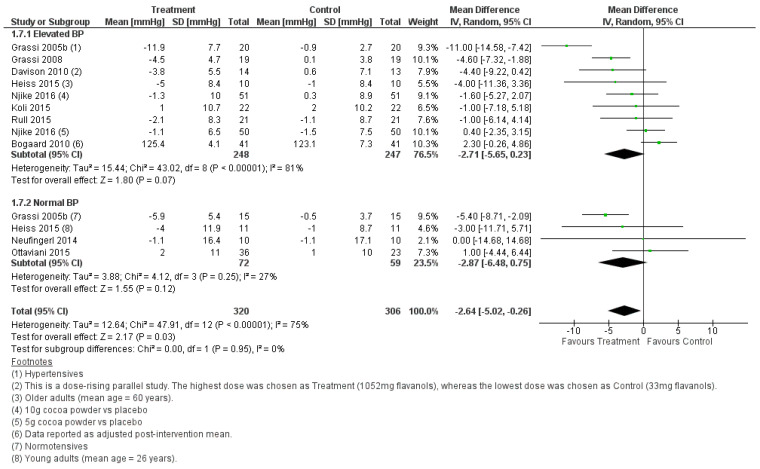
Forest plot showing the pooled effect of cocoa consumption on 24-h systolic BP.

**Figure 7 foods-11-01962-f007:**
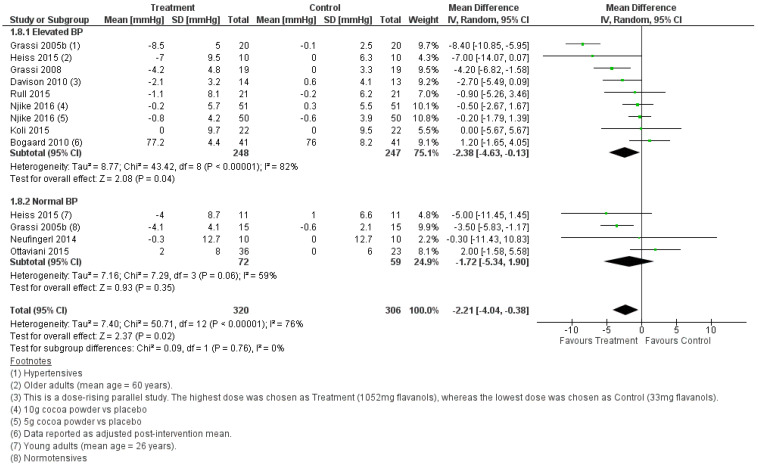
Forest plot showing the pooled effect of cocoa consumption on 24-h diastolic BP.

**Table 1 foods-11-01962-t001:** Summary of study characteristics and BP outcomes listed in alphabetical order by first author surname.

Authors, Year	Country of Study	Age (Years)	BMI (kg/m^2^)	Sample Size	Study Design	Baseline BP	Intervention Duration (Weeks)	Control	Intervention	Serves/Day	Daily Dose of Polyphenol (mg)	Daily Dose of Flavanol (mg)	Daily Dose of Epicatechin (mg)	Outcome
[27]	UK	—	≥25, <25	42	RSBPCT	Normal BP	4	Placebo	Dark chocolate	—	500	—	17	↓ SBP (lean and overweight group) ↓ DBP (overweight group)
[28]	Japan	38	22.1	25	RPT	Normal BP	12	Sugar	Cocoa powder + sugar beverage	2	—	512	377	≠∆ SBP and DBP
[29]	Côte d’Ivoire	18–30	18.5–29.9	124	RSBPT	Normal BP	3	No product	Cocoa beverage	—	590/1475/2950	—	—	↓ SBP (2950 mg polyphenol compared to 590 mg polyphenol)
[30]	USA	≥60	25	90	RDBPCPT	Normal BP	6	Placebo	Chocolate bar and artificially sweetened cocoa beverage	1	—	—	—	≠∆ SBP and DBP
[31]	Australia	53–57	29.3	52	RDBPT	Elevated BP	6	LF cocoa beverage (33 mg flavanol, 0 mg epicatechin)	Cocoa beverage	1	—	372/712/1052	69/138/208	↓ SBP and DBP (1052 mg flavanol compared to 33 mg flavanol)
[32]	Australia	18–65	33.5	49	RDBPCPT	Normal BP	12	LF cocoa drink (36 mg flavanol)	Cocoa beverage	2	—	902	—	↓ DBP (902 mg flavanol compared to 36 mg flavanol) ≠∆ SBP
[33]	USA	32	—	21	RDBPCT	Normal BP	2	Low flavanoid chocolate (trace amount epicatechin)	Chocolate bar	—	—	—	46	≠∆ SBP and DBP
[34]	USA	18	24.1	28	RCT	Normal BP	2	Cocoa butter chocolate	Flavanol-containing milk chocolate	—	—	168	39	↓ DBP and ≠∆ SBP (flavanol-containing milk chocolate) and ≠∆ SBP and DBP (cocoa butter chocolate)
[35]	Spain	57	25.7	140	RSBCPT	Normal BP	26	No product	Chocolate	1	65.5	—	26.1	≠∆ SBP and DBP
[36]	Italy	44.8	<30	19	RSBCT	Elevated BP	2	White chocolate	Chocolate	2	1008	—	110.9	↓ BP (dark but not white chocolate)
[37]	Italy	33.9	22.6	15	RCT	Normal BP	2	White chocolate	Dark Chocolate	—	500	—	—	↓ SBP (dark chocolate compared to white chocolate)
[38]	Italy	43.7	18–27	20	RCT	Elevated BP	2	White chocolate	Dark Chocolate	—	—	88	66	↓ SBP and DBP (dark but not white chocolate)
[39]	Germany	young < 35; elderly 50–80	young 24.9; elderly 26.5	42	RDBPT	Normal BP	2	Placebo	Cocoa beverage	2	—	900	128	Young: ≠∆ SBP, ↓ DBP (treatment compared to control) Elderly: ↓ SBP & DBP (treatment compared to control)
[40]	Finland	45.8	27.7	22	RCCT	Elevated BP	8	Reduced snack intake	Replaced snack intake with dark chocolate	—	—	603	—	≠∆ SBP and DBP
[41]	Spain	25.9	23 (normocholesterolemic); 24 (hypercholesterolemic)	24	NCCFT	Normal BP	4	Milk	Dairy-based cocoa drink	2	—	45.3	18.9	≠∆ SBP and DBP
[42]	USA	24.3	23	40	RDBPCT	Normal BP	4	Placebo tablet	Cocoa tablet	—	—	250	—	≠∆ SBP and DBP
[43]	Australia	40	26 ±4	28	RDBPT	Normal BP	4	Placebo	Cocoa tablet	—	—	234	—	≠∆ SBP and DBP
[44]	USA	51	33.2	20	RDBPCCT	Elevated BP	2	Placebo	Flavanol-rich cocoa drink	2	902	—	174	≠∆ SBP and DBP
[45]	Netherlands	40–70	18.8–30.8	143	RDBPCFFPT	Normal BP	4	Placebo	Cocoa beverage	—	—	325	—	≠∆ SBP and DBP
[46]	Japan	20.2	—	32	RCPT	Normal BP	4	No product	Chocolate	2	—	508	—	≠∆ SBP and DBP
[47]	USA	53.6	<40	122	RDBCMLSPT	Elevated BP	8	Placebo	Chocolate and cocoa beverage	—	—	257/514	24/46	≠∆ SBP and DBP (both high and low dose compared to placebo)
[48]	USA	30–55	24	59	RDBPT	Normal BP	12	Placebo	Cocoa extract capsule	2	—	1000 (week 0–1), followed by 1500 (week 1–3) and 2000 (week 3–12)	110 (week0–1), followed by 165 (week 1–3) and 220 (week 3–12)	≠∆ SBP and DBP
[49]	UK	55.4	26.6	32	RDBPCCT	Elevated BP	6	LF dark chocolate (88 mg flavanol)	HF dark cholate	2	—	1064	—	≠∆ SBP and DBP
[50]	Germany	35–60	23–27	100	RDBCPT	Normal BP	4	Control placebo beverage + theobromine + caffeine	Fruit-flavored cocoa beverage	2	—	900	128	↓ SBP and DBP (cocoa treatment compared to placebo)
[51]	Spain	25–36	<30	24	RCCT	Normal BP	4	Milk	Dairy-based cocoa drink	2	416	—	—	≠∆ SBP and DBP
[52]	Japan	29.7	22.6 ± 2.0/22.6 ± 1.9	39	RSBPT	Normal BP	2	White Chocolate	Dark chocolate	1	550	—	—	≠∆ SBP and DBP
[53]	Indonesia	25–44	18.5–24.9	32	RPT	Elevated BP	2	White Chocolate	Dark chocolate	—	—	—	—	↓ SBP and DBP (dark chocolate not white chocolate)
[54]	Germany	64	>27.5 or <18.5	44	RSBCPT	Elevated BP	18	White Chocolate	Dark chocolate	1	30	—	5.1	↓ SBP and DBP (dark not white chocolate)
[55]	Netherlands	40–70	25.9	41	RDBPCCT	Normal BP	3	Placebo	Flavanol-rich milk-based cocoa drink	—	—	106	—	↑ 24-h ASBP (cocoa flavanol not placebo). ≠∆ PSBP (between cocoa flavanol and placebo after 2 h) ↓ CSBP
[56]	USA	57.7 (HF), 55.4 (LF)	24.9 (HF), 25.3 (LF)	32	RDBPT	Normal BP	6	LF cocoa powder (43 mg flavanol) + sucrose	HF cocoa powder + sucrose	—	—	446	—	↓ SBP and DBP (control but not high dose)
[57]	USA	40–64	25–37	30	RDBPCCT	Normal BP	4	LF chocolate (43 mg flavanol) and sugarless cocoa-free beverage	Dark chocolate + cocoa beverage	—	—	814	—	↑ SBP and DBP (both high dose and control)

Symbols and abbreviations: ↑, high; ↓, low; —, not reported; ≠∆, no change; BP, blood pressure; HF, high-flavanol; LF, low-flavanol; SBP, systolic blood pressure; DBP, diastolic blood pressure; PSBP, peripheral systolic blood pressure; CSBP, central systolic blood pressure; ASBP, ambulatory systolic blood pressure; RSBPCT-Randomized, single-blind, placebo-controlled cross-over trial; RPT-Randomized, parallel trial; RSBPT-Randomized, single-blind parallel trial; RDBPCPT-Randomized double-blind, placebo-controlled, parallel trial; RDBPT-Randomized double-blind, parallel trial; RDBPCPT-Randomized, double-blind, placebo-controlled, parallel trial; RDBPCT-Randomized, double-blind, placebo-controlled trial; RCT-Randomized, cross-over trial; RSBCPT-Randomized, single-blind, controlled parallel trial; RSBCT-Randomized, single-blinded cross-over trial; RCCT-Randomized, controlled, cross-over trial; NCCFT-Non-randomized, controlled, cross-over, free-living trial; RDBPCCT-Randomized, double-blind, placebo-controlled, cross-over trial; RDBPCFFPT-Randomized, double-blind, placebo-controlled, full factorial parallel trial; RCPT-Randomized, controlled, parallel trial; RDBCMLSPT-Randomized, double-blind, controlled, modified Latin square parallel trial; RDBCPT-Randomized, double-blind, controlled, parallel trial; RSBPT-Randomized, single-blind, parallel trial.

**Table 2 foods-11-01962-t002:** Summary of subgroup meta-analysis.

Subgroups	WMD (95% CI) mmHg	*p*-Value	*I*^2^ (%)	*n*	Subgroup Differences (*p*-Value)
Intervention Duration					
Resting Systolic BP					
≤4 weeks	−2.35 [−4.09, −0.61]	0.008	60	22	
>4 weeks	−0.81 [−2.55, 0.92]	0.36	44	12	
Overall	−1.87 [−3.08, −0.65]	0.003	56	34	0.22
Resting Diastolic BP					
≤4 weeks	−2.14 [−3.21, −1.06]	<0.001	35	22	
>4 weeks	−0.38 [−1.52, 0.75]	0.51	43	12	
Overall	−1.38 [−2.18, −0.57]	<0.001	43	36	0.03
24-h Systolic BP					
≤4 weeks	−4.07 [−8.21, 0.07]	0.05	85	7	
>4 weeks	−0.82 [−2.52, 0.87]	0.34	0	6	
Overall	−2.64 [−5.02, −0.26]	0.03	75	13	0.15
24-h Diastolic BP					
≤4 weeks	−4.03 [−6.96, −1.11]	0.007	77	7	
>4 weeks	−1.53 [−0.47, 0.58]	0.38	0	6	
Overall	−2.21 [−4.04, −0.38]	0.02	76	13	0.02 ^a^
Food Matrices					
Resting Systolic BP					
Beverage	−1.54 [−3.08, 0.01]	0.05	0	14	
Chocolate	−3.94 [−5.71, −2.18]	<0.001	63	13	
Beverage + Chocolate	1.22 [−0.86, 3.30]	0.25	0	4	
Tablets/Capsules	1.52 [−3.35, 6.39]	0.54	40	3	
Overall	−1.87 [−3.08, −0.65]	<0.001	56	34	0.002
Resting Diastolic BP					
Beverage	−1.06 [−2.26, 0.15]	0.03	7	14	
Chocolate	−2.59 [−3.78, −1.40]	<0.001	50	13	
Beverage + Chocolate	0.90 [−0.46, 2.26]	0.20	0	4	
Tablets/Capsules	−0.84 [−3.50, 1.83]	0.54	0	3	
Overall	−1.38 [−2.18, −0.57]	<0.001	43	34	0.003
24-h Systolic BP					
Beverage	−1.36 [−5.17, 2.44]	0.48	49	5	
Chocolate	−5.03 [−8.36, −1.70]	0.003	72	5	
Beverage + Chocolate	−0.32 [−2.52, 1.88]	0.78	0	2	
Tablets/Capsules	1.00 [−4.44, 6.44]	0.72	—	1	
Overall	−2.64 [−5.02, −0.26]	0.03	75	13	0.10
24-h Diastolic BP					
Beverage	−2.16 [−5.13, 0.81]	0.15	48	5	
Chocolate	−3.92 [−6.69, −1.15]	0.006	74	5	
Beverage + Chocolate	−0.30 [−1.59, 0.98]	0.64	0	2	
Tablets/Capsules	2.00 [−1.58, 5.58]	0.27	—	1	
Overall	−2.21 [−4.04, −0.38]	0.02	76	13	0.04
Daily Dose of Polyphenols					
Resting Systolic BP					
<500 mg	−0.55 [−3.50, 2.39]	0.71	61	5	
≥500 mg	−2.44 [−4.19, −0.70]	0.006	24	10	
Overall	−1.95 [−3.31, −0.59]	0.005	37	15	0.28
Resting Diastolic BP					
<500 mg	−1.49 [−2.34, −0.65]	<0.001	2	5	
≥500 mg	−1.57 [−2.97, −0.17]	0.03	27	10	
Overall	−1.40 [−2.27, −0.53]	0.002	15	15	0.93
Sensitivity Analysis: Daily Dose of Polyphenols ^b^					
Resting Systolic BP					
<500 mg	0.78 [−2.74, 4.29]	0.66	38	4	
≥500 mg	−2.44 [−4.19, −0.70]	0.006	24	10	
Overall	−1.50 [−3.20, 0.19]	0.08	38	14	0.11
Resting Diastolic BP					
<500 mg	0.24 [−1.49, 2.13]	0.81	0	4	
≥500 mg	−1.57 [−2.97, −0.17]	0.03	27	10	
Overall	−1.14 [−2.25, −0.04]	0.04	16	15	0.14
Daily Dose of Flavanols					
Resting Systolic BP					
<900 mg	−0.39 [−3.95, 3.17]	0.83	79	12	
≥900 mg	−2.88 [−5.06, −0.70]	0.01	0	6	
Overall	−1.31 [−3.72, 1.10]	0.29	69	18	0.24
Resting Diastolic BP					
<900 mg	−0.37 [−2.18, 1.44]	0.69	64	12	
≥900 mg	−2.88 [−4.43, −1.33]	<0.001	0	6	
Overall	−1.22 [−2.58, 0.14]	0.08	56	18	0.04
24-h Systolic BP					
<900 mg	−2.56 [−6.43, 1.31]	0.20	86	7	
≥900 mg	−2.10 [−4.71, 0.52]	0.12	0	5	
Overall	−2.42 [−5.06, 0.22]	0.07	76	12	0.85
24-h Diastolic BP					
<900 mg	−1.94 [−4.68, 0.80]	0.17	85	7	
≥900 mg	−1.94 [−4.71, 0.84]	0.17	49	5	
Overall	−2.01 [−3.97, −0.04]	0.05	77	12	1.00
Sensitivity Analysis: Daily Dose of Flavanols ^c^					
Resting Systolic BP					
<900 mg	1.14 [−0.60, 2.88]	0.20	0	11	
≥900 mg	−2.88 [−5.06, −0.70]	0.01	0	6	
Overall	−0.43 [−1.82, 0.97]	0.55	3	17	0.005
Resting Diastolic BP					
<900 mg	0.43 [−0.67, 1.52]	0.44	0	11	
≥900 mg	−2.88 [−4.43, −1.33]	<0.001	0	6	
Overall	−0.75 [−1.84, 0.34]	0.18	25	17	<0.001
24-h Systolic BP					
<900 mg	0.66 [−0.94, 2.27]	0.42	0	5	
≥900 mg	−2.10 [−4.71, 0.52]	0.12	0	5	
Overall	−0.10 [−1.48, 1.27]	0.88	1	10	0.08
24-h Diastolic BP					
<900 mg	−0.05 [−1.19, 1.09]	0.93	0	5	
≥900 mg	−1.94 [−4.71, 0.84]	0.17	49	5	
Overall	−0.58 [−1.74, 0.59]	0.28	18	10	0.22
Daily Dose of Epicatechin					
Resting Systolic BP					
<100 mg	−1.47 [−4.52, 1.58]	0.35	78	12	
≥100 mg	−3.01 [−4.58, −1.44]	<0.001	0	7	
Overall	−1.89 [−3.81, 0.02]	0.05	66	19	0.38
Resting Diastolic BP					
<100 mg	−0.98 [−2.67, 0.71]	0.26	65	12	
≥100 mg	−2.82 [−4.18, −1.45]	<0.001	0	7	
Overall	−1.53 [−2.76, −0.31]	0.01	55	19	0.10
24-h Systolic BP					
<100 mg	−2.97 [−7.50, 1.56]	0.20	91	5	
≥100 mg	−3.66 [−5.68, −1.63]	<0.001	0	5	
Overall	−2.99 [−5.77, −0.22]	0.03	81	10	0.79
24-h Diastolic BP					
<100 mg	−2.27 [−5.38, 0.84]	0.15	90	5	
≥100 mg	−2.79 [−5.54, −0.04]	0.05	59	5	
Overall	−2.53 [−4.61, −0.45]	0.02	82	10	0.81
Sensitivity Analysis: Daily Dose of Epicatechin ^c^					
Resting Systolic BP					
<100 mg	−0.23 [−1.82, 1.35]	0.77	4	11	
≥100 mg	−3.01 [−4.58, −1.44]	<0.001	0	7	
Overall	−1.42 [−2.66, −0.18]	0.03	13	18	0.01
Resting Diastolic BP					
<100 mg	−0.12 [−1.19, 0.96]	0.83	7	12	
≥100 mg	−2.82 [−4.18, −1.45]	<0.001	0	8	
Overall	−1.13 [−2.17, −0.10]	0.03	30	20	0.002
24-h Systolic BP					
<100 mg	0.66 [−1.43, 2.76]	0.53	34	3	
≥100 mg	−3.66 [−5.68, −1.63]	<0.001	0	5	
Overall	−1.37 [−3.61, 0.87]	0.23	60	8	0.004
24-h Diastolic BP					
<100 mg	−0.05 [−1.22, 1.12]	0.93	0	3	
≥100 mg	−2.79 [−5.54, −0.04]	0.05	59	5	
Overall	−1.31 [−2.98, 0.35]	0.12	60	8	0.07

WMD, weighted mean difference; CI, confidence interval; *I*^2^, heterogeneity; *n*, number of studies. Significance was presented as *p*-value. ^a^ The statistical significance was not robust following sensitivity analysis using the removing-one-treatment-at-a-time method. ^b^ Sensitivity analysis following the removal of Taubert et al. [54] data. ^c^ Sensitivity analysis following the removal of Grassi et al. [38] data.

## Data Availability

Data is contained within the article.

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
