# Peer review of "Effect of Cocoa Beverage and Dark Chocolate Consumption on Blood Pressure in Those with Normal and Elevated Blood Pressure: A Systematic Review and Meta-Analysis"

_foods, 2022, doi:10.3390/foods11131962_

Round 1

Reviewer 1 Report

Overall, the review of Amoah et al is quite interesting and is suitable for publication in Foods. The manuscript is well-organized. I suggest that improve the following points:

1) Table 1 is too extensive (10 pages), mostly due to text in column of "study design". Authors could simply use some abbreviations. There are also several spaces missing between words. Moreover, i suggest that authors highlight the studies in the table by the use of line limits in the table.

2) Please reformat all references in the manuscript according the journal instructons.

Author Response

The authors are very grateful to the reviewer for making time to put together these constructive comments that have led to improving the quality of the manuscript. Thank you so much. Kindly, find below our response to the comments raised for your perusal.

Reviewer 1

Comments and Suggestions for Authors

Overall, the review of Amoah et al is quite interesting and is suitable for publication in Foods. The manuscript is well-organized. I suggest that improve the following points:

Comment: 1) Table 1 is too extensive (10 pages), mostly due to text in column of "study design". Authors could simply use some abbreviations. There are also several spaces missing between words. Moreover, i suggest that authors highlight the studies in the table by the use of line limits in the table.

Response to comment: Thank you for this suggestion. The authors have complied and subsequently used abbreviations for the “study design” as suggested by the reviewer. This can be found in Table 1 and in the footnote of Table 1. This can be found at Line 355-363. The footnote has now got the addition “RSBPCT- Randomized, single-blind, placebo-controlled cross-over trial; RPT- Randomized, parallel trial; RSBPT- Randomized, single-blind parallel trial; RDBPCPT- Randomized double-blind, placebo-controlled, parallel trial; RDBPT- Randomized dou-ble-blind, parallel trial; RDBPCPT- Randomized, double-blind, placebo-controlled, parallel trial; RDBPCT- Randomized, dou-ble-blind, placebo-controlled trial; RCT- Randomized, cross-over trial; RSBCPT-Randomized, single-blind, controlled parallel trial; RSBCT- Randomized, single-blinded cross-over trial; RCCT- Randomized, controlled, cross-over trial; NCCFT- Non-randomized, controlled, crossover, free-living trial; RDBPCCT- Randomized, double-blind, placebo-controlled, crossover trial; RDBPCFFPT- Randomized, double-blind, placebo-controlled, full factorial parallel trial; RCPT- Randomized, controlled, parallel trial; RDBCMLSPT- Randomized, double-blind, controlled, modified Latin square parallel trial; RDBCPT- Randomized, double-blind, controlled, parallel trial; RSBPT- Randomized, single-blind, parallel trial”.

Comment: 2) Please reformat all references in the manuscript according the journal instructions.

Response to comment:

Thank you for this comment. The authors have reformatted all the references using the MDPI referencing style as suggested by the reviewer.

Reviewer 2 Report

The article summarizes the evidence resulting from studies focused on investigating the effects of ≥2 weeks of cocoa consumption as beverage or dark chocolate in those with normal or elevated blood pressure systolic blood pressure measured in the fasted state or over 24h. The effects of different factors and parameters on the impact of cocoa on blood pressure are described. The article is interesting and show a potential effect of cocoa and its components on the blood pressure than could be beneficial to approve health claims related to this food. However, some limitations such as the sample size of the studies included in this meta-analysis that are described in the article, could be responsible for the low effects observed that should be carefully revised studying other parameters, individuals and conditions. Other points that should be addressed the article are the following:

1. The authors should use American or English spelling, but not both of them.

2. Although the article summarizes the human evidence on the effects of cocoa on blood pressure, some information about the potential effects observed by animal models should be included to support the abscence or presence of beneficial effects. It would improve the quality of the introduction. 

3. Lines 158-162: This sentence should not be needed. 

4. Line 165: Once standard deviation has been defined, use the abbreviation SD

5. The authors should carefully revise the journal's guidelines to cite the references into the text. 

6. Table 1: To avoid an excess lenght of the article, this table could be as a supplementary table. 

7. Line 539: The authors should revise the full names and abbreviation. CVD should be used since it was already defined. 

8. Page 16: The references should be reformatted according to the journal's guidelines.

Author Response

The authors are very grateful to the reviewer for making time to put together these constructive comments that have led to improving the quality of the manuscript. Thank you so much. Kindly, find below our response to the comments raised for your perusal.

Reviewer 2

Comments and Suggestions for Authors

The article summarizes the evidence resulting from studies focused on investigating the effects of ≥2 weeks of cocoa consumption as beverage or dark chocolate in those with normal or elevated blood pressure systolic blood pressure measured in the fasted state or over 24h. The effects of different factors and parameters on the impact of cocoa on blood pressure are described. The article is interesting and show a potential effect of cocoa and its components on the blood pressure than could be beneficial to approve health claims related to this food. However, some limitations such as the sample size of the studies included in this meta-analysis that are described in the article, could be responsible for the low effects observed that should be carefully revised studying other parameters, individuals and conditions. Other points that should be addressed the article are the following:

Comment: 1. The authors should use American or English spelling, but not both of them.

Response to comment: Thank you for this comment. The authors have revised the manuscript using American spelling type. For example, in Line 70, “antagonise” has been changed to “antagonize” and in Line 645, “randomised” has been changed to “randomized”.

Comment: 2. Although the article summarizes the human evidence on the effects of cocoa on blood pressure, some information about the potential effects observed by animal models should be included to support the absence or presence of beneficial effects. It would improve the quality of the introduction.

Response to comment:

Thank you for the suggestion. The authors have added a paragraph regarding cocoa intake and its antihypertensive effects in animal models. This can be found at Line 76-88.

Several studies using animal models have established the bioavailablity and antihypertensive effects associated with the intake of cocoa polyphenol. For example, in uninephrectomized hypertensive rats fed for 4 weeks with cocoa feed alone or in combination with an 8% salt diet, the authors reported a significant reductions in both systolic and diastolic BP when compared to uninephrectomized rats fed normal feed [13]. In young spontaneously hypertensive rats, the administration of epicatechin for a two week period resulted in significant reduction in systolic BP [14]. Cienfuegos-Jovellanos et al. [15] administered a single dose of polyphenol-rich cocoa powder to male spontaneously hypertensive rats and reported that up to a dose of 300mg/kg there was a dose-dependent reduction in systolic BP in a manner similar to 50mg/kg of Captopril (a known anti-hypertensive drug). The diastolic BP was maximally reduced over 24 hours by the administration of 50 and 100mg/kg of the polyphenol-rich cocoa powder. Maximum effects were achieved 4 hours post-dose.

Comment: 3. Lines 158-162: This sentence should not be needed.

Response to comment: Thank you for this comment. However, according the Cochrane guideline, Chapter 4, Section 4-5 states that “The search process (including the sources searched, when, by whom, and using which terms) needs to be documented in enough detail throughout the process to ensure that it can be reported correctly in the review, to the extent that all the searches of all the databases are reproducible.” This is the link to the guide: https://training.cochrane.org/handbook/current/chapter-04#section-4-5

Consequently, the requirement by those involved in the search process is essential for transparency in conducting the review. The authors have thus left it as it is to comply to this guideline.

Comment: 4. Line 165: Once standard deviation has been defined, use the abbreviation SD

Response to comment:

Thank you for the reviewers comment. The authors have used the abbreviation SD after expanding the abbreviation in Line 180.

Comment: 5. The authors should carefully revise the journal's guidelines to cite the references into the text.

Response to comment: Thank you for this comment. The authors have reformatted all the references using the MDPI referencing style as suggested by the reviewer.

Comment: 6. Table 1: To avoid an excess length of the article, this table could be as a supplementary table.

Response to comment: Thank you for the comment. The authors have opted against placing it as a “supplementary material” but have used abbreviations for the study design as suggested by the other reviewer which has led towards reducing the bulkiness of Table 1.

Comment: 7. Line 539: The authors should revise the full names and abbreviation. CVD should be used since it was already defined.

Response to comment: Thank you for the comment. The authors have replaced “cardiovascular disease” with “CVD” as suggested. This can be found at Line 710.

Comment: 8. Page 16: The references should be reformatted according to the journal's guidelines.

Response to comment: Thank you for this comment. The authors have reformatted all the references using the MDPI referencing style as suggested by the reviewer.
